# Predictive effect of investor sentiment on current and future returns in emerging equity markets

**Rameeza Andleeb**[ID]*, **Arshad Hassan**

Capital University of Science and Technology, Islamabad, Pakistan

* andleebrameeza@yahoo.com

**Data Availability Statement:** By using the following URLs others can access the datasets: • https://www.investing.com/indices/world-indices • https://stats.oecd.org/ • https://data.imf.org/?sk=

## Abstract

This study uses Non-linear Predictive Regression Analysis to analyze the effect of investor sentiment on the returns of the selected developing equity markets, including Brazil, South Africa, Indonesia, India, China, Russia, and Pakistan. The Principal Component Analysis is applied to construct an Investor Sentiment Index. In most selected countries, investor sentiment substantially affects contemporaneous market returns, and this effect remains persistent in the short term. However, it becomes less prominent over time. It suggests that stakeholders should give importance to the investors' sentiments while making investment decisions.

## Introduction

Standard finance theories proposed by great scholars [1–7] assume that investors are always rational, information is equally disseminated to all investors, prices remain fair in the markets, and consequently, investors cannot gain abnormal profits. The followers of these assumptions claim that these theories provide a simple and understandable explanation of the market phenomena and are applicable to different market conditions. However, these theories still do not give a valid explanation of observed fluctuations in the market asset prices in diverse economic states. To understand the phenomenon, scholars seek help from human psychology from which a new financial paradigm known as behavioral finance takes birth in the field of finance. Behavioral finance assumes that investors are not always rational, information needs to be equally disseminated to all investors, asset prices in the financial market do not stay equitable, and thus investors are able to enjoy abnormal profits.

Keynes's animal spirit theory [8] and Kahneman and Tversky's prospect theory [9] explain the way emotion influences supply and demand. Investor sentiment refers to the overall attitude of buyers and sellers in a specific financial market. Investor sentiment influences supply and demand in the financial market and thus influences the values of financial assets. Market participants determine when to enter and leave the market on the basis of market mood, signs, and trading tactics to maximize their profits. Quick reaction to investor views is critical for maximizing profits in the equity market. The market patterns reflect the buyers' confidence or negativity. Kahneman and Tversky [9] discuss different types of investor irrationality and conclude that investors' cognitive and emotional biases significantly influence their behavior,

4c514d48-b6ba-49ed-8ab9-2b0c1a0179b&sId=
1409151240976.

**Funding:** The authors received no specific funding
for this work.

**Competing interests:** The authors have declared
that no competing interests exist.

leading to few investment choices. This cognitive and emotional imbalance is more frequently seen in investors whose conduct is governed by their feelings and instinct [8]. Commonly, the prevailing market sentiment survives in the short term because it reverts after a certain level.

Shefrin [10] concludes that although various models are in use to investigate behavioral asset pricing theory, still, they need to be more consistent, uniform, and coherent in their attempts to clarify this phenomenon. Behavioral finance theory is recognized as a description of investors' usual stock market behavior and offers an appealing explanation for the unexpected market price behavior seen in the equity markets, despite facing significant criticism. Zhang [11] asserts that the unusual changes in market pricing either are the result of incorrect information or of using accurate information incorrectly. These two scenarios are considered responsible for the origin of fictitious beliefs in irrational investors, and such beliefs are referred to as "investor sentiment."

Consideration of sentiments in predicting stock market returns has been the topic of discussion in the equity markets for a long time, because of the assumption that investor sentiment can be used as a predictor of equity market returns. Dash [12] claims that the existence of a correlation between investor purchasing/selling behavior and market direction is able to forecast returns. Investor sentiment as a predictor of equity returns has been the focus of several studies, involving various proxies, methodologies, data frequencies, investor regimes, and market regimes. A review of the work done in this field reveals that there exists a substantial body of research that claims that investor sentiment is a weak predictor of returns [13–17], but at the same time the number of studies which find sentiment as a reliable predictor of market returns is not less [18–23].

A broad spectrum of literature focuses on the relationship between investor sentiment and equity market returns with reference to time; some find this relationship significant in the long term [24–26], while others find it important in the short term [27–31]. Similarly, investor sentiment is found insignificant predictor of equity market returns in the short-term [32, 33] and in the long-term [14], [30, 34]. When the direction of the association between the predictive power of investor sentiment and returns is focused, contradictory results are observed. Swamy et al. [35] find a positive impact over the short run, Mehrani et al. [31] & Yelamanchili [36], detect a negative relationship in the short term, Liston-Perez et al. [37] see a negative link in the long term and Kadilli [38] report a positive link in the long term.

The existence of non-linearity between investor sentiment and returns was first observed by Nishiyama et al. [39] and is considered responsible for the disagreement observed in the results [40]. Bekiros et al. [40], using $S^{BW}$ and $S^{PLS}$ indices, find no relationship between investor sentiment and market returns under non-linear conditions while Wang [41], using the nonlinear Granger Causality Model, find a non-linear relationship between investor sentiment and equity market returns. Balcilar et al. [42], using the nonparametric Granger Causality Quintile Test and $S^{BW}$ and $S^{PLS}$ indices, confirm investor sentiment as a predictor of stock market returns in nonlinear terms. The contradictory results observed in the literature emphasize the consideration of non-linearity inherent in the data in future studies for the generalization of results. Therefore, the present study investigates the non-linear relationship between investor sentiment and contemporaneous equity returns at the country and group levels. It further explores the importance of investor sentiment in predicting the equity returns for 1, 5, 10, 20, 40, 60, 180, and 240 days at the country as well as group levels.

## Literature review

The study of investor behavior in relation to its impact on stock market returns is valuable in understanding stock markets because the sentiments of the investors are thought to play an

important role in their investment plans, by predicting the returns from their investment. Therefore, investor sentiment and equity returns have remained a hotspot for researchers. Kim et al. [43], by analyzing equity trading data, discovered that changes in investor sentiment predicted the trading behavior of the investors and denoted that during trading, the positive feedback strategy was adopted by individuals, whereas the negative feedback strategy was adopted by institutional investors. Gao & Xie [44] applied the VAR model to daily data and took the CSI 300 index of the future market as a proxy tool to measure investor sentiment. They proved that investor sentiment was a reliable predictor of returns, and this predictor was more sensitive during low margin requirement and high volatility sentiment periods. Lv et al. [45] utilized the textual analysis method and constructed negative and positive investor sentiment indexes using crawler software that captured the data of investor reading volume and post content aggregated from the Chinese stock bar of East Money. They explored the predictive effect of these indices on Chinese equity market returns and demonstrated that negative investor sentiment had a greater predictive effect on equity returns than positive investor sentiment; further, the reaction of investor sentiment towards equity returns in the short term was prominent.

Irrespective of time, no correlation between investor sentiment and returns was observed [13, 15–17, 46, 47]. Whereas, with reference to time, investor sentiment was found an unreliable indicator of stock returns, in the short term [32, 36, 48, 49] as well as in the long term [14, 30, 50, 51]. In terms of directionality, the link between investor sentiment and subsequent market returns in the short term was positive [35, 52–56] as well as negative [31, 57]. Contrarily, in the long term this relationship was positive [58] as well as negative [52, 53, 55, 59].

In global level studies, Ho & Hung [21] selected 4 largest countries in Europe, the United States, and 3 countries of Asia Pacific to study the impact of investor sentiment on returns and revealed that in the United States, France, and Italy, a high level of consumer confidence led to low market returns; and in Japan, shift in consumer confidence increased the market returns in the coming months. Feldman & Liu [60] studied the relationship between expected market returns and investors' sentiment in six developed equity markets and found that during bear periods, the sentiments significantly forecasted stock market return for one year. During financial crises, the impact varied in different markets. Corredor et al. [61] examined the investor sentiment effect on equity returns in four major European markets and found that significant variations in the intensity of sentiment affected the markets, possibly due to differences in stock characteristics and institutional factors. Corredor et al. [62] investigated the predictive effect of investor sentiment on equity returns for Hungary, Poland, and the Czech Republic and found that sentiment had a stronger impact on stock prices in these markets as compared to more developed European markets. Wang et al. [63] examined the role of investor sentiment in explaining stock returns in a global context considering the emerging and developed markets and found that global stock market returns were negatively influenced by investor sentiment in the subsequent 2 to 12 months, however, the impact was more pronounced in developing markets in comparison to developed markets. It became evident from an international study that investor sentiment was an important indicator in the equity markets and that equity returns could best be forecasted when negative returns were also considered [64].

Chitra Devi & Kuppusamy [65] highlighted the limitation of the most commonly used Linear Regression Model and denoted that it satisfied only the linear nature of the relationship between two variables and failed to satisfy the relation of multiple independent variables with dependent variables while forecasting stock values. To overcome this limitation and increase the accuracy of the prediction of returns, a nonlinear regression model was applied to the stock market data, and the result showed that the prediction value was better than that of a linear regression model. Tuyon et al. [66] used various techniques and proxies to measure

investor sentiment under various conditions and found that OLS analysis did not predict the returns significantly. In contrast, QR analysis provided significant asymmetric results, with a U curve pattern, representing positive magnitude at upper and negative magnitude at lower quantiles. The relationship was also significant at extreme quantiles. Lutz [67] found an asymmetric relationship between sentiment and expected future returns; the relationship was strong and negative during sentiment contraction periods (peak-to-trough periods), whereas this relationship was significantly positive but relatively weak during sentiment expansion periods (trough-to-peak). Kabir & Shakur [68] utilized the Smooth Transition Regression (STR) method to analyze herding behavior in Latin American and Asian markets, taking into account different market states and volatility regimes as transition variables and indicated that in high market regimes and high volatile regimes, some markets exhibited herding behavior. This observation differed from previous findings that did not consider nonlinearity in market regimes. In short, the study found herding behavior in certain markets during the high market and high volatile regimes using STR analysis.

Dergiades [69] utilized the investor sentiment index of Baker and Wurgler and explored the nonlinear causal linkage of investor sentiment dynamics and its predictability for US future equity market returns and found a significant predictive power of investor sentiment for US equity returns.

Bekiros et al. [40] utilized the indices of Huang et al. [70] & Baker and Wurgler [71] in the non-linear setting by applying non-parametric regression and found no predictability of investor sentiment for returns of both indexes. Balcilar et al. [42] restudied the work of Bekiros et al. [40] by applying nonparametric quantiles causality test and found that index of Baker & Wurgler, as well as that of Huang et al. [70], predicted the market return in non-linear settings. However, the index of Huang et al. [70] proved a more powerful predictor during bullish and bearish regimes. Balcilar et al. [42] found that linear predictive models showed mixed evidence of predictability for the SBW and SPLS sentiment indices. However, a nonparametric causality-in-quantiles model revealed that both indices could predict stock returns, with SPLS being a better predictor in bearish and bullish regimes. The study highlighted the importance of using non-linear models for regime changes and non-linearity. Xie [72] used nonlinear and linear Granger causality tests to examine a dynamic behavioral finance system which was composed of the willingness of investors towards investment, investor sentiment, and stock index prices and observed that a nonlinear dynamic system predicted the index return better because when nonlinearity existed in the data, using a nonlinear dynamic system lost some explanatory power. Caglar & Ergu [73] observed that the conventional Granger causality test failed to indicate any causality between the investor sentiment index and returns of BIST100.

The discrepancies in the results could be attributed to the investors' innate intellectual ability, education, professional development, emotional stability, the market's proportion of rational and irrational investors, cultural constructs, institutional status, calamities, and information and noise.

## Methodology

The objective is to examine the impact of investor sentiment on current returns and its predictability at various time terms, in both linear and non-linear settings. The countries selected in the sample included Brazil, South Africa, Russia, India, China, Indonesia, and Pakistan because emerging markets are thought more prone to sentiments of investors. The relevant data are collected from the respective markets from 1st January 2001 to 31st December 2020. The macroeconomic data are taken from International Monetary Financial Statistics.

The stock index's daily share prices have been transformed into returns using the following formula:

$$R_t = \ln\left(\frac{P_t}{P_{t-1}}\right) \tag{1}$$

Where ln denotes the natural log, $R_t$ denotes the return of daily prices of the selected index, $P_t$ is the price of the selected index at time t, $P_{t-1}$ represents the price of the selected index at first lag.

The Principal Component Analysis technique of Baker & Wurgler [71, 74] is applied to quantify the investor sentiment index using Trading Volume and Turnover Ratio as proxies.

Investor sentiment index, at the country level, is created by the model:

$$SENT_t = \beta_0 + \beta_1 T_t + \beta_2 R_t + \varepsilon_t \tag{2}$$

Where $SENT_t$ represents the sentiment index at the country level, $T_t$ represents trading volume and $R_t$ represents the turnover ratio.

The Investor sentiment index, at the group level, is constructed by the model:

$$SENT_{i,t} = \beta_0 + \beta_1 T_{i,t} + \beta_2 R_{i,t} + \varepsilon_{i,t} \tag{3}$$

Where $SENT_{i,t}$ denotes sentiment index at the group level, $T_{i,t}$ denotes trading volume, and $R_{i,t}$ denotes the turnover ratio.

To calculate the effect of investor sentiment on current equity returns in linear and nonlinear settings at country and group levels model (4) and model (5) are used respectively.

$$R_{t+0} = \beta_0 + \beta_1 SENT_t^* + \beta_2 SENT_t^{*2} + \beta_3\, Control_t + \beta_4\, AR_t + \varepsilon_t \tag{4}$$

Where $R_{t+0}$ is stock index returns on the current day, $SENT_t^*$ is investor sentiment index calculated from the PCA method, $SENT_t^{*2}$ is the non-linear term, $Control_t$ is controlled macroeconomic variables. Controlled macro-economic variables include; T-Bill, Term Spread, and Industrial Production Index. These variables are supposed as predictors of market returns and thus are frequently used by researchers. $AR_t$ is Auto Regressive term. $\beta_1$ and $\beta_2$ are coefficients that indicate investor sentiment for linear and nonlinear terms respectively.

$$R_{t+0} = \beta_0 + \beta_1 SENT_{i,t}^* + \beta_2 SENT_{i,t}^{*2} + \beta_3\, Control_{i,t} + \beta_4\, AR_{i,t} + \varepsilon_{i,t} \tag{5}$$

Where $R_{t+0}$ is stock index returns on the current day, $SENT_{i,t}^*$ is investor sentiment index calculated from the PCA method, $SENT_{i,t}^{*2}$ is the non-linear term, $Control_{i,t}$ is controlled macroeconomic variables. $AR_{i,t}$ is Auto Regressive term.

To calculate the effect of investor sentiment on current equity returns in linear and nonlinear settings at country and group level model (6) and model (7) are used respectively.

$$R_{t+n} = \beta_0 + \beta_1 SENT_t^* + \beta_2 SENT_t^{*2} + \beta_3\, Control_t + \varepsilon_t \tag{6}$$

Where $R_{t+n}$ is the stock index return at the country level at a time $(t + n)$. For each country the value of n is 1,5,10 and 20 days for the short term; 40,60,180 and 240 days for the long term as used by Kim [50].

$$R_{t+n} = \beta_0 + \beta_1 SENT_{i,t}^* + \beta_2 SENT_{i,t}^{*2} + \beta_3\, Control_{i,t} + \varepsilon_{i,t} \tag{7}$$

Where $R_{t+n}$ is stock index return at the time $(t + n)$. The value of n is 1,5,10 and 20 days for the short term; 40,60,180 and 240 days for the long term as used by Kim [50].

## Results and discussion

In order to examine the impact of investor sentiment on current-day returns, Linear and Non-Linear Regressive Models are applied at the country level, and Dynamic Panel Data Analysis is at the group level in the first phase. Then Predictive Linear and Predictive Non-Linear Regressive Models are employed at the country level and Predictive Dynamic Panel Data Technique at the group level in the second phase. Auto-Regressive Terms are used where Auto-correlation exists in the series.

Table 1 shows the descriptive statistics of the variables studied for country and group levels.

The acronyms used in Table 1 are RTP (equity market returns), SENTIM (Investor Sentiment Index), TS (Term Spread), TB (Risk-free rate), and IPI (Industrial Production Index). Daily market equity returns range from 0.02 (South Africa) to 0.07 (Pakistan). The variability of returns ranges from 20.204 to -7.741, being a minimum (of 8.255 to -7.74) for Pakistan and maximum (of 20.204 to -21.199) for Russia. The values of standard deviation are the lowest (0.007) for Pakistan and the highest (2.036) for Russia. The values of Skewnes are negative and the values of Kurtosis are greater than 3. The values of the Investor Sentiment Index are negative for Pakistan and Russia only. The value of the standard deviation of the sentiment index is lowest (1.304) in South Africa and highest (1.395) in India. The Investor Sentiment Index is negatively skewed only for South Africa and kurtosis values are greater than three indicating an abnormality in the data.

The average T-bills and Industrial Production Index values for all the selected countries are positive. The values of the standard deviation of T-Bills range from 1.22 (India) to 5.19 (Brazil). The value of Skewnes is positive for all the sample countries except Pakistan and the value of Kurtosis is in the range of 3. The variability in the values of the standard deviation of the Industrial Production Index ranges from 24.39 to 5.33 for all the sample countries. Average Term Spread values are negative only for Pakitan and Brazil and positive for the rest of the countries. Pakistan shows the lowest (0.03) and Brazil shows the highest (3.65) value of the standard deviation of Term Spread. The Skewnes value of Term Spread is positive for Indonesia, India, China, and Pakistan and negative for the rest of the countries and groups. The values of Kurtosis are more than 3.

Table 2 shows the effect of investor sentiments and macro factors on same-day returns for country and group levels.

In Brazil, China, Indonesia, Pakistan, and Russia investor sentiment affects same-day equity returns significantly positively. This relationship is nonlinear in all the sample countries except India and South Africa. The negative signs observed in the non-linear term indicate the convexity in the relationship and correction of mispricing at a certain higher level. The linear and negative relationship observed in the case of India and South Africa indicates that an increase in the sentiment of investors decreases the equity returns. This negative influence is also observed by Brown & Cliff [75]. Considering the data at the panel level it becomes evident that current-day equity returns are significantly and positively affected by investor sentiment in linear settings. This positive influence indicates that a high level of investor sentiment gives high returns. These results are in accordance with those of Lux [76] and Changsheng & Yongfeng [77]. In non-linear settings, the relationship between investor sentiment and returns is insignificant in India and at the panel level and these results support the results of Bekiros et al. [40].

Table 3 shows the predictive effect of investor sentiment on equity returns on days 1, 5, 10, 20, 40, 60, 180, and 240. A time period of less than one month is taken as short term whereas time from 1 to 12 months is taken as a long term.

**Table 1. Descriptive statistics.**

| | BRAZIL | RUSSIA | INDONESIA | INDIA | CHINA | SOUTH AFRICA | PAKISTAN | PANEL |
|---|---|---|---|---|---|---|---|---|
| | | | | Market Returns (RTP) | | | | |
| Mean | 0.0552 | 0.0392 | 0.0483 | 0.0560 | 0.0395 | 0.0271 | 0.0758 | 0.0486 |
| Median | 0.1019 | 0.1253 | 0.1117 | 0.0976 | 0.0779 | 0.0640 | 0.1053 | 0.0969 |
| Maximum | 13.6766 | 20.2039 | 7.6234 | 16.3343 | 9.4008 | 9.0570 | 8.2547 | 20.2039 |
| Minimum | -15.9930 | -21.1994 | -10.9539 | -13.9038 | -9.2562 | -10.4504 | -7.7414 | -21.1994 |
| Std. Dev. | 1.7739 | 2.0360 | 1.3077 | 1.4115 | 1.5067 | 1.3154 | 1.2922 | 1.5428 |
| Skewness | -0.3864 | -0.5947 | -0.9657 | -0.5248 | -0.3624 | -0.1258 | -0.4917 | -0.5231 |
| Kurtosis | 10.1148 | 14.1198 | 11.7092 | 14.3874 | 8.1124 | 7.9781 | 6.4138 | 13.0506 |
| | | | | Sentiment index (SENTIM) | | | | |
| Mean | -0.0002 | -0.0007 | 0.0000 | 0.0146 | 0.0000 | 0.0000 | -0.0183 | -0.0004 |
| Median | -0.7583 | 0.3182 | 0.0004 | -0.1631 | 0.0461 | 0.1169 | -0.1954 | -0.0757 |
| Maximum | 5.8453 | 6.5330 | 7.6657 | 5.6438 | 5.9152 | 9.5081 | 7.4606 | 9.5081 |
| Minimum | -1.3048 | -2.4861 | -4.6454 | -5.3668 | -2.8464 | -3.5152 | -5.9747 | -5.9747 |
| Std. Dev. | 1.3516 | 1.3613 | 1.3492 | 1.3947 | 1.3439 | 1.3042 | 1.3310 | 1.3482 |
| Skewness | 1.3920 | 0.3030 | 0.7548 | 1.1240 | 0.5384 | -0.5610 | 0.4055 | 0.5901 |
| Kurtosis | 4.6646 | 2.3486 | 5.1726 | 4.7595 | 3.7478 | 7.5907 | 6.8124 | 4.9479 |
| | | | | Treasury bill rates (TB) | | | | |
| Mean | 12.2554 | 8.4611 | 8.2104 | 6.6705 | 1.0310 | 7.4251 | 8.5154 | 7.5094 |
| Median | 11.7515 | 7.6555 | 7.1804 | 6.0000 | 0.3000 | 7.1150 | 8.8100 | 7.0300 |
| Maximum | 28.7800 | 27.8300 | 17.3900 | 10.2500 | 4.7600 | 12.7400 | 14.0100 | 28.7800 |
| Minimum | 0.0000 | 0.0000 | 4.0600 | 4.2500 | -0.0800 | 3.4500 | 1.2100 | -0.0800 |
| Std. Dev. | 5.1943 | 3.5506 | 3.1043 | 1.2264 | 1.2589 | 2.0240 | 3.3123 | 4.3912 |
| Skewness | 0.2408 | 1.6252 | 1.3162 | 0.9567 | 1.2385 | 0.7401 | -0.2633 | 0.7102 |
| Kurtosis | 3.1246 | 8.2539 | 4.0252 | 3.4356 | 3.2957 | 3.0792 | 2.3962 | 4.5016 |
| | | | | Term Spread (TS) | | | | |
| Mean | -0.2203 | 0.8143 | 1.4614 | 0.7909 | 1.0286 | 0.5060 | -0.0100 | 0.6272 |
| Median | 0.1660 | 0.9500 | 1.1995 | 0.6240 | 0.9400 | 1.1500 | -0.0097 | 0.7130 |
| Maximum | 5.7500 | 7.1120 | 6.5880 | 3.2320 | 3.9500 | 5.1350 | 0.1249 | 7.1120 |
| Minimum | -25.0575 | -7.3900 | -4.2760 | -1.8970 | -0.8380 | -89.1740 | -0.1300 | -25.0575 |
| Std. Dev. | 3.6521 | 2.4841 | 0.9726 | 0.7780 | 0.4262 | 2.6401 | 0.0333 | 2.0255 |
| Skewness | -5.3347 | -1.2450 | 0.8187 | 0.8566 | 1.0522 | -9.0400 | 0.6080 | -5.7652 |
| Kurtosis | 35.4485 | 6.0778 | 3.4766 | 3.7975 | 4.4071 | 261.4086 | 10.9703 | 66.0694 |
| | | | | Industrial Production Index (IPI) | | | | |
| Mean | 97.0220 | 134.5883 | 104.2800 | 81.8216 | 97.8604 | 111.2959 | 97.0220 | 103.4117 |
| Median | 99.9027 | 137.7056 | 97.9985 | 80.9702 | 98.0083 | 111.5000 | 99.9027 | 102.0304 |
| Maximum | 135.5511 | 181.4030 | 149.0540 | 123.4824 | 125.1021 | 135.1089 | 135.5511 | 181.4030 |
| Minimum | 36.0068 | 83.3553 | 55.0730 | 3.4697 | 79.7375 | 92.9173 | 36.0068 | 3.4697 |
| Std. Dev. | 16.9459 | 24.3976 | 16.6906 | 18.2289 | 5.3321 | 6.0488 | 16.9459 | 22.2400 |
| Skewness | -0.4430 | -0.1415 | 0.4735 | -0.6246 | 0.4429 | 0.2744 | -0.4430 | 0.4732 |
| Kurtosis | 2.4819 | 2.1370 | 2.5857 | 4.4048 | 3.4468 | 3.4006 | 2.4819 | 4.5546 |
| Observations | 5218 | 5218 | 5218 | 5218 | 5218 | 5218 | 5218 | 36533 |

Table 3 shows that in Brazil the relationship of investor sentiment with market returns in linear settings is significantly negative at days 1 and 5 and significantly positive on days 60 and 240. This means that at earlier stages increase in investor sentiment decreases the returns, under the influence of selling pressure in the market, and this effect is gradually reversed at later stages. In non-linear settings, the relationship is significantly positive on days 1 and 5 and

**Table 2. Effect of investor sentiment on current day equity returns.**

| Variable | Brazil | Russia | Indonesia | India | China | South Africa | Pakistan | Panel |
|---|---|---|---|---|---|---|---|---|
| CONSTANT | -0.464805 | 1.579540*** | 0.806359** | 0.471007** | −4.266229*** | -0.154717 | 0.343593* | 0.160914*** |
| SENTIM | 0.204895* | 0.073643* | 0.183050*** | −0.045581*** | 0.384214*** | −0.040018** | 0.386019*** | 0.026633*** |
| SENTIM$^2$ | −0.045126* | −0.067299*** | −0.011975* | 0.002939 | −0.060503*** | 0.003665 | 0.028399*** | -0.000933 |
| TS | 0.000861 | −0.044433* | 0.023118 | −0.000690* | 0.242169*** | -0.004849 | 2.016163*** | -0.004655 |
| TB | -0.007968 | −0.032430** | 0.028715** | -0.021977 | 0.187816*** | −0.046354*** | 0.036436*** | -0.000728 |
| IPI | 0.004764 | −0.007840** | −0.009357*** | −0.003343** | 0.041013*** | 0.004767 | −0.005553*** | −0.001042*** |
| AR(1) | 0.085782*** | −0.764479*** | −0.495966*** | 0.130393*** | −0.895273*** | −0.358148*** | −0.659972*** | 0.115118*** |
| *Adj R*2 | 0.1 | 3 | 4.5 | 1.7 | 2 | 1.1 | 8.4 | 1.4 |
| D-W | 1.945646 | 1.999769 | 1.999945 | 2.003279 | 1.999085 | 1.997589 | 1.996081 | 2.005229 |

The acronyms used in Table 2 are SENTIM (Investor Sentiment Index), TS (Term Spread), TB (Risk-free rate), IPI (Industrial Production Index), and AR (Auto Regressive term).

* is significant at 1%.

** is significant at 5% and

*** denotes significance at 10%.

becomes significantly negative at days 40 and 60. It means that mispricing observed in the short term is corrected over the long term. In linear settings and short term, a positive effect is observed by Chang et al. [52], whereas a negative effect is observed by Da et al. [78]. Similarly in non-linear settings and in the long run, positive effect is observed by Fang et al. [58], and the negative effect is observed by Chang et al. [52].

In Russia, investor sentiment affects the returns significantly in linear settings while in non-linear terms the effect is significant and positive at days 1, 10, 40, and 240 and convex in nature. The results are in line with those of Cheema et al. [79]. In Indonesia, the effect of sentiment in linear terms is less pronounced as it predicted the returns only for the next day and at day 180 whereas in nonlinear terms it predicted the returns at day 40 and 60 only. Similar results are noted by Yelamanchili [36] and Kling & Gao, [14]. In the Chinese market, the relation of investor sentiment with returns in linear settings is positive on days 1, 5, 10, 40, and 60 and in nonlinear settings it is negative at days 1, 5, 10, 40, 60, and 180, and the nature of the relationship is convex in nature which means sentiment is increasing the returns at decreasing rate. In India, investor sentiment has a significant nonlinear impact on returns on all days except day 60. In South Africa, it has a significant effect at day 40 only in linear terms and at days 1,5, and 180 in nonlinear terms. The positive signs indicate the existence of mispricing in the market. The Pakistani market exhibits a significant positive relationship in both linear and nonlinear terms which indicates that sentiment is increasing the returns at an increasing rate.

At the group level linear relationship between investor sentiments and future market, returns are observed on days 1, 5, 10, and 20. The nonlinear relationship is observed after 10, 20, and 40 days. These results line up with the study of Ruan et al. [80] and Fang et al. [81]. Term Spread do not impact returns in sample countries individually as well as at the group level. Industrial Production Index shows a positive relation to equity returns, which indicates that with an increase in growth rate returns become higher.

Risk-Free Rate has an insignificant effect on equity returns in the markets of Brazil, Indonesia, Pakistan, and Russia, at various time terms, and a significant effect in China, South Africa, and India. The controversy in the context of direction may be due to divergent economic situations and varying degrees of control in each country. At the group level, this variable does not affect returns. Similar nature of results is observed with the Term Spread variable. Industrial

**Table 3. Predictive effect of investor sentiment on equity returns at various time horizons.**

| Investor Sentiments and Market Returns -Brazil | | | | | | | |
|---|---|---|---|---|---|---|---|
| **Short term** | | | | **Long term** | | | |
| Variable | $R_{t+1}$ | $R_{t+5}$ | $R_{t+10}$ | $R_{t+20}$ | $R_{t+40}$ | $R_{t+60}$ | $R_{t+180}$ | $R_{t+240}$ |

| Variable | $R_{t+1}$ | $R_{t+5}$ | $R_{t+10}$ | $R_{t+20}$ | $R_{t+40}$ | $R_{t+60}$ | $R_{t+180}$ | $R_{t+240}$ |
|---|---|---|---|---|---|---|---|---|
| CONSTANT | −0.622754*** | 0.836602* | 0.088666 | 0.049209 | 0.115650 | 0.125437 | 0.225402 | 0.178840 |
| SENTIM | −0.160632*** | −0.126118* | 0.020531 | 0.007539 | 0.024972 | 0.054753* | 0.036798 | 0.063976* |
| SENTIM$^2$ | 0.046177* | 0.039777*** | -0.009822 | -0.010535 | −0.024734*** | −0.023425*** | -0.002876 | -0.002743 |
| TS | 0.013610 | -0.003549 | -0.008674 | −0.012428* | −0.011613* | -0.007072 | −0.013444* | -0.004321 |
| TB | -0.010817 | 0.005403 | 0.003297 | 0.007628 | 0.005727 | -0.002296 | -0.000826 | −0.009477* |
| IPI | 0.006831*** | −0.007277* | -0.000614 | -0.000758 | -0.000913 | 5.70E-07 | -0.001569 | 3.86E-05 |
| *Adj R2* | 0.028 | 0.005 | 0.001 | 0.002 | 0.003 | 0.002 | 0.002 | 0.002 |
| D-W | 2.444781 | 1.713432 | 1.970089 | 1.971209 | 1.971879 | 1.970856 | 1.969798 | 1.9794 |
| **Investor Sentiments and Market Returns -Russia** | | | | | | | | |
| CONSTANT | 0.123382 | 0.156853** | 0.034118 | -0.018271 | −0.115994* | -0.075221 | 0.025015 | -0.059309 |
| SENTIM | −0.109287*** | −0.053387** | −0.059444*** | −0.046821* | −0.047267* | −0.045154* | 0.039018* | −0.047310* |
| SENTIM$^2$ | 0.037697*** | 0.010693 | 0.044779*** | 0.014699 | 0.033754** | 0.013587 | 0.018137 | 0.045949*** |
| TS | −0.046263*** | −0.022339* | -0.009753 | -0.001745 | -0.002160 | -0.003755 | 0.013803 | -0.005905 |
| TB | −0.035588*** | −0.014053** | -0.008367 | 0.003202 | 0.010744* | 0.010102 | 0.003311 | 0.001089 |
| *Adj R2* | 0.029 | 0.026 | 0.029 | 0.024 | 0.027 | 0.026 | 0.0005 | 0.027 |
| D-W | 1.998700 | 1.997695 | 2.001632 | 1.993152 | 1.993585 | 1.994249 | 1.683372 | 1.999413 |
| **Investor Sentiments and Market Returns -Indonesia** | | | | | | | | |
| CONSTANT | 0.336155* | 0.382218* | 0.374791* | 0.256437 | 0.177479 | 0.224008 | 0.259363 | 0.232205 |
| SENTIM | 0.039387* | 0.002615 | 0.017445 | 0.001556 | 0.023771 | 0.012577 | 0.045686* | 0.002998 |
| SENTIM$^2$ | 0.004670 | 0.006173 | 0.004868 | 0.002646 | 0.013427* | 0.012061* | 0.000332 | 0.002347 |
| TB | 0.004370 | 0.008506 | 0.000586 | 0.003373 | 0.006730 | 0.003898 | 0.004820 | 0.005883 |
| TS | 0.019758 | 0.004998 | 0.020842 | 0.000150 | 0.007587 | 0.016960 | 0.011712 | 0.008871 |
| IPI | 0.003330** | 0.002573* | 0.003461** | 0.001778 | 0.001042 | 0.001319 | 0.002390 | 0.002159 |
| AR (1) | 0.068378*** | 0.192371*** | 0.192163*** | 0.193236*** | 0.193593*** | 0.191329*** | 0.191411*** | 0.191706*** |
| *Adj R2* | 0.005 | 0.039 | 0.04 | 0.039 | 0.04 | 0.039 | 0.039 | 0.039 |
| D-W | 1.630171 | 1.999372 | 1.999293 | 1.999376 | 1.999462 | 1.999189 | 1.999477 | 1.999201 |
| **Investor Sentiments and Market Returns -India** | | | | | | | | |
| CONSTANT | -0.027315 | 0.840594*** | 0.771072*** | -0.390756 | 0.684602*** | −0.619058* | -0.240051 | 0.348827 |
| SENTIM | 0.120486 | 0.002449 | -0.014381 | −0.194701** | 0.007064 | 0.009996 | -0.022582 | 0.152472* |
| SENTIM$^2$ | −0.047557*** | −0.014173*** | −0.010995* | 0.048211*** | −0.017120*** | 0.005501 | 0.026514* | −0.023695* |
| TS | 0.182740*** | 0.041560* | 0.037281* | 0.149576*** | 0.037240* | 0.148187*** | 0.029843 | −0.064365* |
| TB | 0.090757* | −0.084872*** | −0.069666** | 0.130871*** | −0.057485* | 0.099141** | 0.025055 | −0.069139* |
| IPI | −0.009367*** | −0.002831* | −0.003201** | −0.005730*** | −0.002818** | -0.002062 | 0.000785 | 0.001319 |
| *Adj R2* | 0.036 | 0.012 | 0.002 | 0.012 | 0.003 | 0.006 | 0.009 | 0.004 |
| D-W | 1.972589 | 1.759697 | 1.769311 | 1.986917 | 1.762261 | 1.685769 | 2.007870 | 1.961537 |
| **Investor Sentiments and Market Returns -China** | | | | | | | | |
| CONSTANT | -0.508671 | -0.376836 | -0.363796 | 0.179582 | -0.073682 | 0.226249 | -0.034639 | 0.278439 |
| SENTIM | 0.047760*** | 0.043803** | 0.047542* | 0.028908 | 0.043946* | 0.036207* | 0.006713 | -0.012840 |
| SENTIM$^2$ | −0.015853*** | −0.021475*** | −0.023926*** | 0.030439*** | −0.023667*** | −0.017203** | −0.014437** | -0.006155 |
| TS | 0.005504 | 0.037341 | 0.043194 | 0.000770 | 0.103496* | 0.110351* | 0.022410 | -0.008325 |
| TB | 0.052880*** | 0.066213*** | 0.069134*** | 0.063533*** | 0.059709*** | 0.058943*** | 0.037110** | 0.028810* |
| IPI | 0.005200 | 0.003560 | 0.003389 | 0.001538 | -0.000140 | -0.003400 | 0.000387 | -0.002544 |
| AR(1) | 0.192490*** | 0.036401*** | 0.782514*** | 0.788380*** | 0.785593*** | 0.787428*** | 0.064021*** | 0.805747*** |
| *Adj R2* | 0.013 | 0.008 | 0.01 | 0.01 | 0.0009 | 0.009 | 0.004 | 0.008 |
| D-W | 1.991523 | 1.997833 | 1.993697 | 1.992977 | 1.993637 | 1.992974 | 2.001834 | 1.996693 |

(*Continued*)

**Table 3.** (Continued)

| | Investor Sentiments and Market Returns -Brazil | | | | | | | |
|---|---|---|---|---|---|---|---|---|
| | Short term | | | | Long term | | | |
| Variable | $R_{t+1}$ | $R_{t+5}$ | $R_{t+10}$ | $R_{t+20}$ | $R_{t+40}$ | $R_{t+60}$ | $R_{t+180}$ | $R_{t+240}$ |
| Investor Sentiments and Market Returns- South Africa | | | | | | | | |
| CONSTANT | 0.236933 | −0.780548* | 0.176787 | 0.277281 | 0.453814* | 0.551859* | 0.159813 | 0.037277 |
| ΔSENTIM | 0.023511 | 0.033998 | 0.013561 | 0.008067 | −0.034448* | -0.023424 | -0.022972 | 0.018434 |
| ΔSENTIM$^2$ | 0.014483** | 0.014252** | -0.000399 | 0.003136 | 0.001589 | 0.004430 | 0.014973* | 0.005696 |
| TS | -0.001457 | 0.020484 | -0.005311 | 0.000890 | -0.008970 | -0.003536 | 0.010368* | -0.001326 |
| TB | −0.026160*** | -0.005404 | −0.028066*** | −0.029300*** | −0.028821*** | −0.024238*** | -0.009205 | -0.007899 |
| IPI | -0.000213 | 0.007234* | 0.000553 | -0.000320 | -0.001886 | -0.003102 | -0.000666 | 0.000437 |
| AR(1) | 0.052525*** | −0.102042*** | −0.017743** | −0.020531** | 0.996514*** | 0.996338*** | 0.988283*** | 1.001840*** |
| *Adj R*2 | 0.011 | 0.016 | 0.011 | 0.012 | 0.012 | 0.011 | 0.012 | 0.009 |
| D-W | 1.997595 | 1.859148 | 1.999737 | 1.987652 | 1.990963 | 1.988252 | 1.982433 | 1.988703 |
| Investor Sentiments and Market Returns -Pakistan | | | | | | | | |
| CONSTANT | 0.199576* | 0.193348 | 0.227494* | 0.190085* | 0.196475 | 0.136581 | 0.118795 | 0.118004 |
| SENTIM | 0.042032*** | 0.044933*** | 0.053462*** | 0.008476 | 0.022258* | 0.009693 | 0.009644 | 0.010524 |
| SENTIM$^2$ | 0.000723 | 0.009208** | 0.018769*** | 0.023798*** | 0.005893* | 0.007207** | 0.007677 | 0.009162* |
| TS | 0.892403* | 0.027055 | 0.659842 | 1.586349*** | 0.012867 | 0.241910 | 0.940325* | 0.251002 |
| TB | 0.007754 | 0.007918 | 0.003660 | 0.010133* | 0.006968 | 0.007208 | 0.006993 | 0.001771 |
| IPI | 0.000609 | 0.000336 | 0.000946 | 5.32E-07 | 0.000505 | 0.000192 | 0.000170 | 0.000645 |
| AR(1) | 0.075068*** | 0.179942*** | 0.178467*** | 0.186288*** | 0.562588*** | 0.564061*** | 0.571824*** | 0.974568*** |
| *Adj R*2 | 0.013 | 0.043 | 0.045 | 0.042 | 0.042 | 0.042 | 0.042 | 0.039 |
| D-W | 1.659891 | 2.001380 | 2.001351 | 2.019094 | 1.988736 | 1.988028 | 1.987092 | 2.001152 |
| Panel Data Results | | | | | | | | |
| CONSTANT | 0.151879*** | 0.170923*** | 0.192050*** | 0.182481*** | 0.173481*** | 0.142672*** | 0.114700** | 0.127630*** |
| SENTIM | 0.012355** | 0.014042** | 0.015073** | 0.011687* | 0.005208 | 0.001180 | 0.002248 | 0.000768 |
| SENTIM$^2$ | 0.001407 | 0.000780 | 0.004997** | 0.007247*** | 0.005145* | 0.002521 | 0.003087 | 0.003298 |
| TS | 0.003714 | 0.001862 | 0.001944 | 0.001452 | 0.005265 | 0.003503 | 0.003680 | 0.005784 |
| TB | 0.001329 | 0.000560 | 0.000200 | 0.000940 | 0.001048 | 0.001420 | 0.002092 | 0.002243 |
| IPI | 0.000937*** | 0.001130*** | 0.001317*** | 0.001247*** | 0.001179*** | 0.000951** | 0.000802* | 0.000918* |
| AR (1) | 0.125315*** | 0.032914*** | 0.017994*** | 0.123187*** | 0.120573*** | 0.120301*** | 0.118475*** | 0.118270*** |
| *Adj R*2 | 0.016 | 0.017 | 0.016 | 0.015 | 0.016 | 0.016 | 0.015 | 0.015 |
| D-W | 1.999943 | 2.000565 | 2.000314 | 2.007224 | 1.999944 | 2.000475 | 1.998940 | 2.000310 |

The acronyms used in Table 2 are SENTIM (Investor Sentiment Index), TS (Term Spread), TB (Risk-free rate), and IPI (Industrial Production Index), and AR (Auto Regressive term).

* is significant at 1%.

** is significant at 5% and

*** denotes significance at 10%.

Production Index has a positive impact on the returns of Brazil, South Africa, and Indonesia, no impact on the returns of Pakistan and China, negative impact on the returns of India.

## Conclusion and recommendations

In both linear and nonlinear settings, current-day returns are significantly influenced by investor sentiment in all the countries under study except India, South Africa, and Panel where it is significant only in linear settings. Investor sentiment is predicting the equity returns in all the

countries and even at the panel level in both linear and non-linear settings most of the time terms. In short and linear terms the direction of the relationship between the two is positive for China and Pakistan and negative for Russia. In long and linear terms this relationship is negative for Indonesia and Russia and positive for Pakistan. In non-linear terms, the direction of the relationship is positive for Russia, South Africa, Pakistan, and Panel in both short and long terms whereas it is negative for China and India most of the time terms. The positive effect indicates a continuation of mispricing and the negative effect indicates the correction of mispricing. In the case of Brazil, the relationship is positive in the short term that is reversed in the long term. The reversal of coefficient signs indicates that asset mispricing observed in the short term is corrected over time.

The nutshell of the study is that investor sentiment is a predictor of equity returns in both linear and nonlinear terms at various times terms. Policy-makers may keep these findings in mind while devising investment policies in the financial equity markets. These findings may be helpful for individual investors and portfolio managers in understanding the market trends and the role of sentiments in market functioning. Risk managers can also make more accurate predictions and devise appropriate strategies for better allocation of assets by considering the sentiment of investors in the equity market.

Low values of $Adj\ R^2$ observed in this study were also observed in the studies of Kim [50] and Chang & Luo [82], indicating that investor sentiment is not the only factor that influences equity returns, a lot of other factors also have their effect on market functioning that should also be considered in the asset pricing models in future studies.

## Author Contributions

**Conceptualization:** Rameeza Andleeb, Arshad Hassan.

**Data curation:** Rameeza Andleeb, Arshad Hassan.

**Formal analysis:** Rameeza Andleeb, Arshad Hassan.

**Investigation:** Rameeza Andleeb.

**Methodology:** Rameeza Andleeb, Arshad Hassan.

**Resources:** Rameeza Andleeb.

**Software:** Rameeza Andleeb.

**Supervision:** Arshad Hassan.

**Validation:** Rameeza Andleeb, Arshad Hassan.

**Visualization:** Rameeza Andleeb.

**Writing – original draft:** Rameeza Andleeb.

**Writing – review & editing:** Rameeza Andleeb.

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
