## [Decision Letter · Decision Letter 0]

30 Mar 2023

PONE-D-23-02126Impact of Investor Sentiment on Contemporaneous and Future Equity Returns in Emerging MarketsPLOS ONE

Dear Dr. ANDLEEB,

Thank you for submitting your manuscript to PLOS ONE. After careful consideration, we feel that it has merit but does not fully meet PLOS ONE’s publication criteria as it currently stands. Therefore, we invite you to submit a revised version of the manuscript that addresses the points raised during the review process.

We look forward to receiving your revised manuscript.

Kind regards,

Saqib Farid

Academic Editor

PLOS ONE

Additional Editor Comments:

The paper tackles an interesting topical research question. It would surely benefit from the following comments and suggestions that are aimed to make it better crafted and more polished.

Reviewer # 1

Dear Authors,

Hope this mail finds you well.

After examining your submission I have decided not to seek the advice of reviewers. Your paper entitled “Impact of Investor Sentiments on Contemporaneous and Future Equity Returns in Emerging Markets “offers a set of very interesting aspects. While your findings illustrate merit, however, the manuscript can be improved in few ways before publication.

the Main points are:

1. The authors should better highlight the research gap and motivation of the study.

2. The contribution of the papers should be clearly explained.

3. The literature review section should discuss few latest papers on the topic.

4. The choice and rational to justify the use of specific methods needs to be improved.

5. The discussion on the results requires to be expanded, where focus should be paid on corroborating the findings with previous evidence, and economic interpretation of the results should be improved.

Good look with the revision!

Reviewer # 2

Thank you very much for giving me this opportunity to review the manuscript entitled “ Impact of Investor Sentiments on Contemporaneous and Future Equity Returns in Emerging Markets “ After carefully reading the manuscript, I have few suggestions for the author(s) to incorporate in the manuscript and improve its quality further.

Comment 1: I can see that there is contribution of study but is not well-written or explicitly

stating what novelty is being added in the study.

Comment 2: What is the rationale behind using underlying methods in the current research?

Why the authors did not employ other similar techniques and preferred this technique solely?

Comment 3: Are there any other studies on the similar methodology? If yes, please cite them

to have a better empirical justification.

Comment 4: What are future prospects of this study? Adding the future directions along with

the implications for individual market would be greater advantage of the study.

Comment 5: I found some grammatical and typo errors in the manuscript. In that case, a

professional proofread can polish and enhance the quality of your manuscript.

Good Luck with the revision !

Reviewers' comments:

Reviewer's Responses to Questions

**Comments to the Author**

1. Is the manuscript technically sound, and do the data support the conclusions?

Reviewer #1: Yes

Reviewer #2: Yes

2. Has the statistical analysis been performed appropriately and rigorously? 

Reviewer #1: Yes

Reviewer #2: Yes

3. Have the authors made all data underlying the findings in their manuscript fully available?

Reviewer #1: Yes

Reviewer #2: Yes

4. Is the manuscript presented in an intelligible fashion and written in standard English?

Reviewer #1: Yes

Reviewer #2: Yes

5. Review Comments to the Author

Reviewer #1: Dear Authors,

Hope this mail finds you well.

After examining your submission I have decided not to seek the advice of reviewers. Your paper entitled “Impact of Investor Sentiments on Contemporaneous and Future Equity Returns in Emerging Markets “offers a set of very interesting aspects. While your findings illustrate merit, however, the manuscript can be improved in few ways before publication.

the Main points are:

1. The authors should better highlight the research gap and motivation of the study.

2. The contribution of the papers should be clearly explained.

3. The literature review section should discuss few latest papers on the topic.

4. The choice and rational to justify the use of specific methods needs to be improved.

5. The discussion on the results requires to be expanded, where focus should be paid on corroborating the findings with previous evidence, and economic interpretation of the results should be improved.

Good look with the revision!

Reviewer #2: Thank you very much for giving me this opportunity to review the manuscript entitled “ Impact

of Investor Sentiments on Contemporaneous and Future Equity Returns in Emerging Markets “

After carefully reading the manuscript, I have few suggestions for the author(s) to incorporate

in the manuscript and improve its quality further.

Comment 1: I can see that there is contribution of study but is not well-written or explicitly

stating what novelty is being added in the study.

Comment 2: What is the rationale behind using underlying methods in the current research?

Why the authors did not employ other similar techniques and preferred this technique solely?

Comment 3: Are there any other studies on the similar methodology? If yes, please cite them

to have a better empirical justification.

Comment 4: What are future prospects of this study? Adding the future directions along with

the implications for individual market would be greater advantage of the study.

Comment 5: I found some grammatical and typo errors in the manuscript. In that case, a

professional proofread can polish and enhance the quality of your manuscript.

Good Luck with the revision !

6. PLOS authors have the option to publish the peer review history of their article (what does this mean?). If published, this will include your full peer review and any attached files.

Reviewer #1: No

Reviewer #2: No

---

## [Author Response · Author response to Decision Letter 0]

16 Apr 2023

Reviewer 1

First of all, thank you to review the research paper and for your valuable suggestions. All the suggestions have been incorporated, which improved the document. The revised paper is being submitted with a request to kindly consider the same for publication.

The suggested changes have been incorporated with detail as follows:

Comments to the Author: The authors should better highlight the research gap and motivation of the study.

Author Response: The Research Gap is revised in the manuscript as per the suggestions.

Comment to the Author: The contribution of the papers should be clearly explained.

Author Response: The contribution of the papers is clearly explained in the last section of the manuscript.

Comment to the Author: The literature review section should discuss a few latest papers on the topic.

Author Response: Change has been incorporated into the document (The latest studies added in the document)

Comment to the Author: The choice and rationale to justify the use of specific methods needs to be improved.

Author Response: The choice and rationale is revised and improved by justifying the use of specific methods.

Comment to the Author: The discussion on the results requires to be expanded, where the focus should be paid to corroborating the findings with previous evidence, and economic interpretation of the results should be improved.

Author Response: The discussion on the results is expanded by supporting the findings, and the economic interpretation of the results is also improved.

Reviewer 2

First of all, thank you to review the research paper and for your valuable suggestions. All the suggestions have been incorporated, which improved the document. The revised paper is being submitted with a request to kindly consider the same for publication.

The suggested changes have been incorporated with detail as follows:

Comment to the Author: I can see that there is a contribution of the study but is not well-written or explicitly states what novelty is being added to the study.

Author Response: The contribution of the study is revised that explicitly explain the way ‘novelty’ is being added to the study.

Comment to the Author: What is the rationale behind using underlying methods in the current research? Why the authors did not employ other similar techniques and preferred this technique solely?

Author Response: The choice and rationale of the underlying method is revised and improved by justifying the use of a specific technique.

Comment to the Author: Are there any other studies on a similar methodology? If yes, please cite them to have a better empirical justification.

Author Response: yes, there are several studies available on a similar methodology, in text citations and references are inserted at the appropriate place.

Comment to the Author: What are the future prospects of this study? Adding the future directions along with the implications for the individual market would be the greater advantage of the study.

Author Response: The Future Prospects, Future Directions, and Implications in the conclusion and recommendation section have been added. 

Comment to the Author: I found some grammatical and typo errors in the manuscript. In that case, a professional proofreader can polish and enhance the quality of your manuscript.

Author Response: The document is checked properly through grammar. All typo errors have been removed and all suggested changes in the entire manuscript are done.

Regards

Authors

---

## [Editor Report · Decision Letter 1]

28 Apr 2023

Predictive Effect of Investor Sentiment on Current and Future Returns in Emerging Equity Markets

PONE-D-23-02126R1

Dear Dr. ANDLEEB,

We’re pleased to inform you that your manuscript has been judged scientifically suitable for publication and will be formally accepted for publication once it meets all outstanding technical requirements.

Kind regards,

Saqib Farid

Academic Editor

PLOS ONE
---

## [Editor Report · Acceptance letter]

9 May 2023

PONE-D-23-02126R1 

Predictive Effect of Investor Sentiment on Current and Future Returns in Emerging Equity Markets 

Dear Dr. Andleeb:

I'm pleased to inform you that your manuscript has been deemed suitable for publication in PLOS ONE. Congratulations! Your manuscript is now with our production department. 

Kind regards, 

on behalf of

Dr. Saqib Farid 

Academic Editor

PLOS ONE